# Association between obstructive sleep apnea risk and atherosclerosis: A nationwide cross-sectional study in the Korean population

Il Rae Park , Yong Geun Chung, Seung Min Chung *, Jun Sung Moon,
Ji Sung Yoon, Kyu Chang Won

Department of Internal Medicine, Yeungnam University College of Medicine, Daegu

* smchung@ynu.ac.kr

## Abstract

### Objectives

Obstructive sleep apnea (OSA) increases the risk of all-cause and cardiovascular mortality. This study aimed to investigate the association between OSA and atherogenic risk in the Koreans.

### Methods

Data from 8,158 participants (mean age, 57.9 ± 11.7; male/female, 1:1.4) obtained from the Korea National Health and Nutrition Examination Survey between 2019 and 2021. OSA risk was screened using the STOP-BANG score, and atherogenic risk was measured using the atherogenic index of plasma (AIP). Logistic regression was used to evaluate the association between the STOP-BANG scores and high AIP and subgroups according to the presence of diabetes.

### Results

The proportions of individuals with atherogenic risk (AIP > 0.24) were 13.7%, 27.6%, and 34.7% in the low-, intermediate-, and high-OSA risk groups ($p < 0.001$). After adjustment, individuals with intermediate and high OSA risk had 1.35 (95% confidence interval [CI], 1.16–1.58; $p < 0.001$) and 1.32 (95% CI, 1.08–1.61; $p = 0.006$) times higher odds of having atherogenic risk than those with low OSA risk. Among patients without diabetes, high OSA risk was not an independent factor affecting atherogenic risk (hazard ratio [HR], 1.17; 95% CI, 0.93–1.47). However, among patients with diabetes, compared with those with low OSA risk, those with intermediate (HR, 1.51; 95% CI, 1.05–2.19) and high OSA risk (HR, 1.58; 95% CI, 1.02–2.46) had significantly increased atherogenic risk.

**Data availability statement:** Access to the raw Korea Health and Nutritional Examination Survey(KNHANES) data is open to the public; however, users must submit a signed "Statistical Data User Compliance Agreement" and "Confidentiality Agreement" to ensure responsible data use. The data underlying the results presented in the study are available from the KNHANES official website. (https://knhanes. kdca.go.kr/knhanes/sub03/sub03_02_05.do).

**Funding:** This work was supported by the New Faculty Research Grant of Yeungnam University (224A580066), awarded to Ilrae Park. The funders had no role in study design, data collection and analysis, decision to publish, or preparation of the manuscript. URL: https:// www.yu.ac.kr

**Competing interests:** The authors have declared that no competing interests exist.

## Conclusion

OSA is linked to increased atherogenic risk in the Koreans, especially in individuals with diabetes, thus highlighting the importance of routine OSA screening to manage and reduce cardiovascular risks.

## Introduction

Obstructive sleep apnea (OSA) is characterized by recurrent partial or total obstruction of the upper airway during sleep that leads to apnea. OSA is a widespread sleep disorder, affecting approximately 936 million adults aged 30–69 years worldwide [1]. Furthermore, with the increasing prevalence of obesity and metabolic syndrome, the incidence of OSA is increasing.

OSA is associated with cardiovascular diseases (CVDs), cerebrovascular diseases, cognitive disorders, type 2 diabetes, and metabolic syndrome [2]. OSA shares similar pathophysiological features with metabolic disorders. The intermittent hypoxia during apnea, which is caused by upper airway obstruction, leads to tissue hypoxia, triggering a series of adverse physiological responses. These include increased sympathetic nervous system activity, oxidative stress, and systemic inflammation, which all contribute to coronary atherosclerosis, insulin resistance, and other metabolic dysfunctions [3]. Consequently, OSA is a significant independent risk factor for CVD [3,4]. A previous meta-analysis of 16 studies involving 24,308 individuals showed that severe OSA increases all-cause and cardiovascular mortality [5].

However, despite the risk associated with the disorder, it is frequently underdiagnosed owing to its occurrence during sleep, which limits patient awareness [6]. The gold standard for diagnosing OSA is polysomnography. However, polysomnography is complex, expensive, and lengthy, limiting its feasibility for widespread use [7]. The STOP-BANG score, created in 2008, is a simple and practical tool that assesses OSA risk through a questionnaire [8]. The questionnaire includes parameters such as snoring, daytime tiredness, observed apnea during sleeping, hypertension, body mass index (BMI) >35 kg/m$^2$, age > 50 years, neck circumference >40 cm, and male sex [8]. A prospective study involving 435 hospitalized patients showed that a STOP-BANG score ≥5 is an independent risk factor for cardiovascular mortality [9]. However, OSA risk is rarely assessed in diabetes care settings.

The atherogenic index of plasma (AIP) can be easily calculated based on lipid profile. It is a well-known marker related to insulin resistance, type 2 diabetes, and CVD and is considered to have better predictive accuracy than individual lipid profiles or glucose alone [10,11]. Furthermore, the AIP is known to be correlated with sleep disorders and the apnea-hypopnea index in previous studies. However, most of these studies have been small-scale cross-sectional studies [12], and none have been conducted on the general population in Korea.

This study aimed to evaluate OSA risk using the STOP-BANG scoring system and to explore its correlation with atherosclerosis risk, which is calculated by AIP. Through

this research, we hope to highlight the broader atherosclerotic and metabolic implications of OSA and reinforce the importance of effective screening and management strategies.

## Methods

### Study participants

This is a cross-sectional study based on data from the Korea National Health and Nutrition Examination Survey (KNHANES) conducted between 2019 and 2021. The survey included 8,110 participants in 2019, 7,359 participants in 2020, and 7,090 participants in 2021, resulting in a total sample size of 22,559 participants. This trial was approved by institutional review board of Yeungnam University Hospital (2024-07-034). In addition, we utilized fully anonymized data from the KNHANES. The study authors did not have access to any information that could identify individual participants during or after study. For this reason, the requirement for informed consent was waived by the IRB of Yeungnam University Hospital.

Participants with a measurable STOP-BANG score were included in the study, which accounted for 11,423 individuals. Exclusion criteria, which were applied to ensure the reliability and relevance of the data, were participants who were unable to calculate the AIP (n = 242), had a history of cerebrovascular accident (n = 342), had a history of myocardial infarction (n = 179), had aspartate aminotransferase (AST) levels ≥200 or alanine aminotransferase (ALT) levels ≥175 (n = 18), had liver cirrhosis (n = 52), had current cancer (n = 290), had a glomerular filtration rate ≤15 mL/min/1.73 m$^2$ (n = 10), or were currently taking medication for dyslipidemia (n = 2,573). The current study investigated the correlation between the STOP-BANG score and the AIP. To ensure the accuracy of the results, we excluded conditions that could affect cholesterol levels, such as severe liver disease, which affects lipid profile due to alteration in hepatic synthesis of cholesterol [13,14]; end-stage renal disease, which alters cholesterol metabolism and causes impaired high-density lipoprotein cholesterol (HDL-C) function [15,16]; additionally, cancer and its treatment can significantly alter patient's physiology, including their immune system, lipid metabolism, and overall health [17]. After these exclusions, 8,158 participants were included in the final analysis (Fig 1).

### Clinical and laboratory measurements

Data collection involved demographic information such as age and sex, clinical measurements including BMI and blood pressure, and laboratory tests such as lipid profiles and liver enzymes. In the KNHANES project, test reliability was ensured through ongoing monitoring of the diagnostic laboratory, including inspections, quality control, inter-laboratory comparisons, traceability checks, and duplicate testing.[18] Lifestyle factors such as current smoking, high-risk alcohol consumption, aerobic exercise, and resistance exercise, as well as socioeconomic factors such as education and household income, were included. Current smokers are defined as people who have smoked more than five packs (100 cigarettes) in their lifetime and are currently smoking. High-risk alcohol consumption is defined as people who drink, on average, more than seven glasses at one time for men and more than five glasses for women and drink more than two times a week. Aerobic exercise is defined as engaging in at least 2 h and 30 min of moderate-intensity physical activity per week, at least 1 h and 15 min of vigorous-intensity physical activity per week, or a combination of moderate and vigorous-intensity activities (with 1 min of vigorous-intensity activity being equivalent to 2 min of moderate-intensity activity) to meet the recommended amount of physical activity. Resistance exercise is defined as individuals who have engaged in resistance exercise exercises such as push-ups, sit-ups, dumbbells, barbells, or pull-ups for at least 2 d in the past week.[18] The education variable was categorized into four groups according to the highest level of education attained: elementary school or lower, middle school graduate, high school graduate, college graduate, or higher. The household income was defined by household income quartiles.

The laboratory tests included fasting glucose (mg/dL), glycated hemoglobin (HbA1c) (%), total cholesterol (mg/dL), HDL-C (mg/dL), triglycerides (TG, mg/dL), and calculated LDL-C (total cholesterol − [TG/5 + HDL-C]), which was used

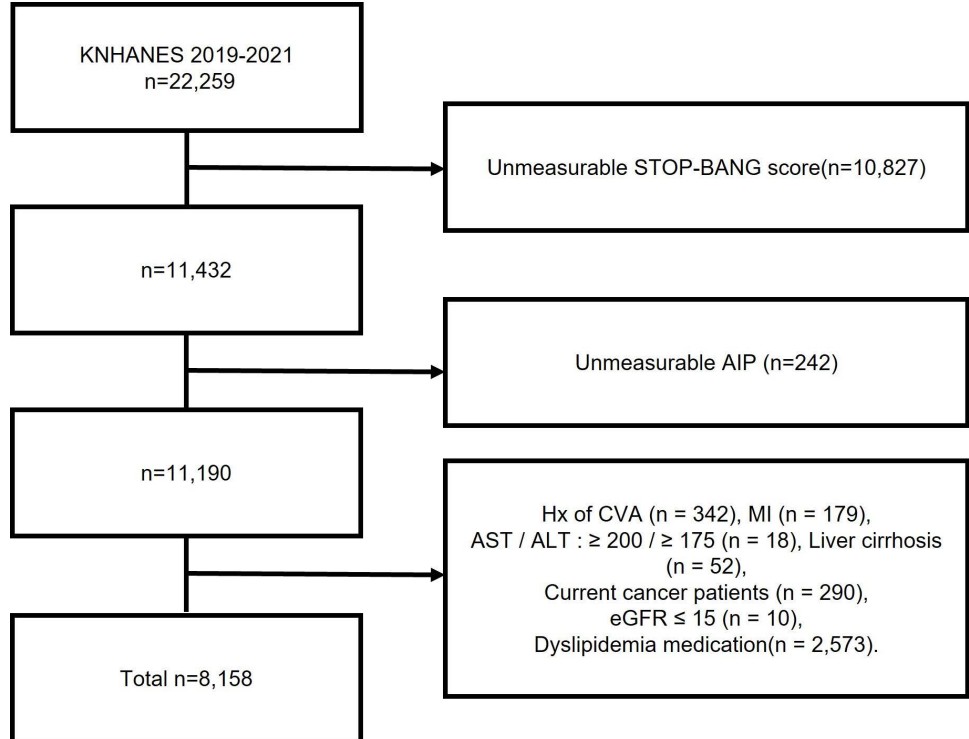

**Fig. 1. Patient flowchart.**

instead of direct LDL-C because of many missing values. AST (IU/mL), ALT (IU/mL), blood urea nitrogen (mg/dL), and creatine (mg/dL) were included. The presence of diabetes in patients was classified based on HbA1c levels, fasting glucose levels, use of diabetes medication, and physician diagnosis. Normal is defined as having a fasting glucose level <100 mg/dL or an HbA1c ≤ 5.7%. Prediabetes is defined as having a fasting glucose level between 100 and 125 mg/dL or an HbA1c between 5.7% and 6.4%. Diabetes is defined as having a fasting glucose level ≥126 mg/dL, taking oral hypoglycemic agents, receiving insulin injections, having a previous doctor's diagnosis, or having an HbA1c ≥ 6.5% [18]. The homeostatic model assessment for insulin resistance (HOMA-IR) and Homeostatic Model Assessment for β-cell function (HOMA-β) were calculated using the following formulas:

$$HOMA-IR = fasting insulin\,(\mu U/mL) \times fasting glucose\,(mg/dL)\,/405$$

and

$$HOMA-\beta = 20 \times fasting insulin\,(\mu U/mL)\,/fasting glucose\,(mg/dL) - 3.5.$$

## Assessment of OSA

The STOP-BANG score, a validated screening tool for OSA, comprises eight questions that assess risk factors related to OSA. Each item is scored as either 0 or 1, with a total possible score ranging from 0 to 8 [8]. The components of the STOP-BANG questionnaire are the following:

  S: snoring loudly (louder than talking or loud enough to be heard through closed doors)

T: tiredness, i.e., feeling tired, fatigued, or sleepy during the daytime

O: observed apnea, i.e., observed patients stop breathing while sleeping

P: high blood pressure

B: BMI > 35 kg/m² [8]. However, a lower BMI threshold should be set for the Asian population. In a study involving 1,205 individuals of different ethnicities, the optimal area under the receiver operating characteristic curve value for BMI to diagnose high-risk OSA was lower for Chinese and Indian individuals [19]. In this study, we set the threshold at 30 kg/m².

A: age > 50 years

N: neck circumference >40 cm

G: gender (1 = men; 0 = women) [8].

Participants were classified based on their STOP-BANG scores into three risk OSA categories: low risk (0–2 points), intermediate risk (3–4 points), and high risk (≥5 points). Additionally, participants in the intermediate-risk category were reclassified as high-risk if they scored ≥2 on the STOP portion and were male, had a BMI ≥ 30, or had a neck circumference >40 cm [8].

The reliability of the STOP-BANG score has been published elsewhere. With a stepwise increase in the STOP-BANG score from 3 to 8, the probability of severe OSA increases from 15% to 75% in the general population [20], and a score >5 is reliable for detecting severe OSA with 90% to 100% sensitivity [8,21,22]. The STOP-BANG questionnaire effectively screens for OSA, with a sensitivity of up to 93.9% and an area under the curve of 0.86 in patients with cardiovascular risk factors [23].

### Assessment of atherogenic risk

We calculated the AIP for the assessment of atherosclerosis and metabolic dysfunction. The AIP is a logarithmic transformation of the ratio of TGs to HDL-C (log[TG/HDL-C]), and an AIP value >0.24 is considered high risk for cardiovascular events [12,24,25].

### Statistical analyses

We analyzed categorical variables using a chi-square test and expressed the data as percentages. Continuous variables were analyzed with an analysis of variance to compare across the groups, and the results were shown as means with a standard deviation. Tukey's HSD post-hoc analysis was conducted to compare differences between the groups. For variables that did not meet the assumptions of normality or homogeneity of variance, the use of non-parametric tests was considered reasonable to ensure robust and reliable analysis.

To evaluate the atherogenic risk associated with low, intermediate, and high STOP-BANG scores, a logistic regression analysis was conducted. This approach allowed us to calculate the odds ratios (ORs) for high AIP with high and intermediate STOP-BANG scores relative to those with low STOP-BANG scores. Covariates such as sex, age, smoking status, alcohol consumption, education, income, aerobic exercise, resistance exercise, diabetes mellitus, HbA1c, and HOMA-IR were included in the logistic regression models to control for potential confounding factors. Model 1 was adjusted for sex and age. Model 2 was adjusted for model 1 + smoking, high-risk alcohol consumption, education, house income, aerobic exercise, and resistance exercise. Model 3 was adjusted for model 2 + diabetes, HbA1c, and HOMA-IR. The results of these analyses provided adjusted odds ratios, indicating the strength of the association between high STOP-BANG scores and elevated metabolic risks, independent of these confounders. Statistical significance was assessed with two-tailed

tests and has a cutoff p-value <0.05. All statistical analyses were performed using R for Windows, version 4.4 (The R Project, Vienna, Austria).

## Results

### Baseline characteristics

Table 1 provides the clinical characteristics of the study population, which includes 8,158 participants (male, n = 3,389, 41.5%; female, n = 4,769, 58.5%; average age, 57.9 years). Participants were categorized by OSA risk based on the STOP-BANG questionnaire. Specifically, 4,681 participants (57.4%) were classified as having low risk, 2,421 (29.7%) as having intermediate risk, and 1,056 (12.9%) as having high risk. The high-OSA risk group had a higher proportion of male participants, elevated blood pressure, and a higher BMI than the low- and intermediate-OSA risk groups. Additionally, the high-OSA risk group had higher rates of current smokers and high-risk alcohol drinkers. This group had a higher prevalence of diabetes and elevated fasting glucose and HOMA-IR levels. In terms of lipid profiles, the high-risk group had higher TG levels and lower HDL-C levels.

The AIP values were −0.09 ± 0.30, 0.06 ± 0.31, and 0.13 ± 0.32 for the low-, intermediate-, and high-OSA risk groups, respectively (p < 0.001). Fig 2 shows the proportion of atherogenic risk (AIP > 0.24) according to the STOP-BANG score. In the high-OSA risk group, 34.7% were at atherogenic risk, which was significantly higher than that of the intermediate- and low-risk groups (27.6% and 13.7%, respectively; p < 0.001).

### Effect of OSA on atherogenic risk

The effect of OSA on atherogenic risk (AIP > 0.24) was explored using logistic regression analysis (Table 2). An unadjusted model showed that, compared with the low-OSA risk group, the high-OSA risk group had a significantly greater risk of having AIP > 0.24 (OR, 3.34; 95% CI, 2.86–3.88; p < 0.001). The intermediate-OSA risk group showed a significantly greater risk than the low-OSA risk group (OR, 2.40; 95% CI, 2.12–2.71; p < 0.001). Even after adjusting for confounding variables such as sex, age, sociological factors (household income, education), lifestyle factors (smoking, alcohol consumption, exercise), and comorbidities such as diabetes, the high-OSA risk group still showed a significant association with AIP > 0.24 (OR, 1.42; 95% CI, 1.16–1.73; p < 0.001), as did the intermediate-OSA risk group (OR, 1.53; 95% CI, 1.30–1.79; p < 0.001). The presence of diabetes was significantly associated with an increased risk of AIP > 0.24; compared with normoglycemic participants, the ORs for AIP > 0.24 were 1.69 (95% CI, 1.46–1.96) and 1.71 (95% CI, 1.32–2.23) for those with prediabetes and diabetes, respectively (Table 3).

### Effect of OSA on atherogenic risk according to the presence of diabetes

Patients were classified as non-diabetic (normal and prediabetes) and diabetic, and the association between atherogenic risk according to OSA risk groups was further analyzed. We examined fasting glucose, HbA1c, HOMA-IR, and AIP among participants with and without diabetes according to OSA risk, and the results are represented in Fig 3&4. The levels of fasting glucose and HbA1c did not differ according to OSA risks regardless of the presence of diabetes (Fig 3A, 3B and 4A, 4B). However, HOMA-IR and AIP levels and the proportion of atherogenic risk had an increasing tendency with statistical significance according to OSA risks among participants with and without diabetes (Fig 3C, 3D, 3E and 4C, 4D, 4E).

The effect of OSA on atherogenic risk among participants with and without diabetes is discussed in Table 4. Logistic regression analysis models were adjusted for covariates considered in Table 2, except for the presence of diabetes. After multivariate adjustments, among participants without diabetes, intermediate (OR, 1.21; 95% CI, 1.02–1.45) but not high OSA risk (OR, 1.17; 95% CI, 0.93–1.47) was an independent factor for atherogenic risk. Meanwhile, among participants with diabetes, intermediate (OR, 1.51; 95% CI, 1.05–2.19) and high OSA risk (OR, 1.58; 95% CI, 1.02–2.46) significantly increased the risk of atherogenic risk.

**Table 1. Clinical characteristics.**

| | | STOP BANG score | | | P-value |
|---|---|---|---|---|---|
| | Total (N = 8,158) | Low (N = 4,681) | Intermediate (N = 2,421) | High (N = 1,056) | |
| Age(yr) | 57.9±11.7 | 55.7±11.6 | 61.8±11.0 | 58.2±11.4 | <0.001 |
| Sex | | | | | <0.001 |
| Male | 3389(41.5) | 729 (15.6) | 1716 (70.9) | 944 (89.4) | |
| Female | 4769(58.5) | 3952 (84.4) | 705 (29.1) | 112 (10.6) | |
| SBP(mmHg) | 121.1±16.7 | 117.0±15.7 | 126.1±16.7 | 127.6±15.6 | <0.001 |
| DBP(mmHg) | 76.0±9.9 | 74.1±8.9 | 77.6±9.9 | 81.1±11.1 | <0.001 |
| BMI(kg/m$^2$) | 23.8±3.4 | 23.0±3.0 | 24.6±3.2 | 26.1±3.9 | <0.001 |
| Current Smoker | 1248 (15.3) | 415(8.9) | 542(22.4) | 291(27.6) | <0.001 |
| High risk Alcohol consumption | 846 (10.4) | 278(5.9) | 334(13.8) | 234(22.2) | <0.001 |
| Aerobic exercise | 3190 (39.1) | 1849 (39.5) | 944 (39.0) | 397 (37.6) | 0.5139 |
| Resistance training | 1699 (20.8) | 853 (18.2) | 597 (24.7) | 249 (23.6) | <0.001 |
| House Income | | | | | <0.001 |
| 1quartile | 1566 (19.3) | 779 (16.7) | 587 (24.3) | 200 (19.0) | |
| 2quartile | 1991 (24.5) | 1143 (24.5) | 576 (23.9) | 272 (25.8) | |
| 3quartile | 2159 (26.6) | 1280 (27.5) | 615 (25.5) | 264 (25.0) | |
| 4quartile | 2406 (29.6) | 1455 (31.2) | 633 (26.3) | 318 (30.2) | |
| Education | | | | | <0.001 |
| Elemental school or lower | 1676 (20.6) | 870 (18.6) | 600 (24.8) | 206 (19.5) | |
| Middle school | 909 (11.1) | 469 (10.0) | 322 (13.3) | 118 (11.2) | |
| High school | 2738 (33.6) | 1611 (34.4) | 788 (32.6) | 339 (32.1) | |
| College graduate or higher | 2830 (34.7) | 1727 (36.9) | 710 (29.3) | 393 (37.2) | |
| Prediabetes | 3914 (49.1) | 2143(46.7) | 1234(52.4) | 537(52.1) | <0.001 |
| Diabetes | 1078(13.5) | 387(8.4) | 445(18.9) | 246(23.9) | <0.001 |
| Fasting Glucose(mg/dl) | 102.5±22.6 | 98.9±19.8 | 106.6±24.8 | 108.7±25.7 | <0.001 |
| HbA1c(%) | 5.8±0.8 | 5.7±0.7 | 6.0±0.9 | 6.0±0.9 | <0.001 |
| HOMA-IR | 2.2±2.4 | 1.9±1.7 | 2.5±2.9 | 3.0±3.1 | <0.001 |
| HOMA-β | 81.4±74.3 | 79.0±78.8 | 81.2±63.0 | 92.7±77.2 | <0.001 |
| Total Cholesterol(mg/dl) | 200.7±38.0 | 203.8±36.7 | 196.1±38.4 | 197.7±40.8 | <0.001 |
| HDL-Cholesterol(mg/dl) | 52.1±12.9 | 55.1±13.0 | 48.6±11.7 | 46.8±11.2 | <0.001 |
| Triglyceride(mg/dl) | 133.8±109.3 | 116.0±80.0 | 150.2±124.2 | 175.1±157.8 | <0.001 |
| LDL-Cholesterol(mg/dl) | 121.9±35.6 | 125.4±33.5 | 117.6±36.5 | 116.1±40.3 | <0.001 |
| AST(IU/ml) | 24.6±10.9 | 23.2±9.0 | 26.2±12.6 | 27.5±13.1 | <0.001 |
| ALT(IU/ml) | 22.1±14.5 | 19.3±11.8 | 24.6±15.5 | 29.0±19.3 | <0.001 |
| BUN(mg/dl) | 15.5±4.6 | 14.8±4.4 | 16.4±4.7 | 16.4±4.9 | <0.001 |
| CRE(mg/dl) | 0.8±0.2 | 0.7±0.1 | 0.9±0.2 | 0.9±0.2 | <0.001 |
| eGFR(mL/min/1.73m$^2$) | 95.9±13.9 | 98.6±12.9 | 92.0±14.0 | 92.7±15.2 | <0.001 |
| AIP | -0.02±0.31 | -0.09±0.3 | 0.06±0.31 | 0.13±0.32 | <0.001 |

OSA, obstructive sleep apnea; SBP, systolic blood pressure; DBP, diastolic blood pressure; BMI, body mass index; HOMA-IR, Homeostatic Model Assessment for Insulin Resistance; HOMA- β, homeostatic model assessment of β-cell function; HDL, high-density lipoprotein; LDL, low-density lipoprotein; AST, aspartate aminotransferase; ALT, alanine aminotransferase; BUN, blood urea nitrogen; CRE, creatinine; eGFR, estimated glomerular filtration rate; AIP, atherogenic index of plasma; AIP calculated by logarithm[Triglyceride/HDL-C]

Total

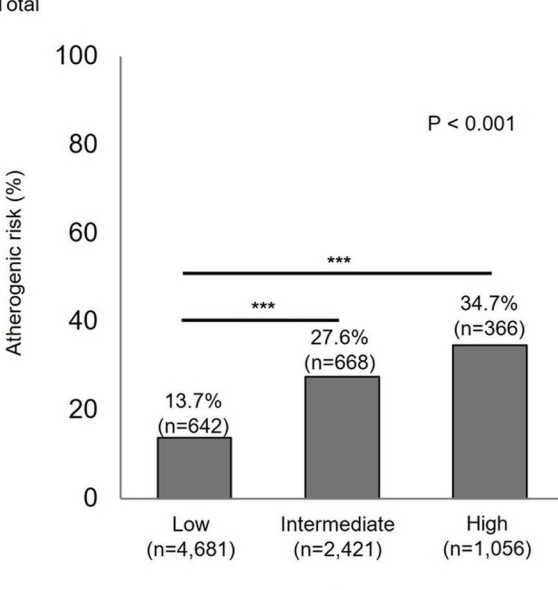

**Fig 2. Proportion of atherogenic risk (AIP > 0.24) according to OSA risk.** Comparison of intermediate or high OSA risk to low OSA risk: ***p < 0.001. AIP, atherogenic index of plasma; OSA, obstructive sleep apnea.

**Table 2. Effect of OSA on atherogenic risk (AIP > 0.24).**

|  | Crude | | Model 1 | | Model 2 | | Model 3 | |
|---|---|---|---|---|---|---|---|---|
|  | OR(95% CI) | p | OR(95% CI) | p | OR(95% CI) | p | OR(95% CI) | p |
| OSA risk |  |  |  |  |  |  |  |  |
| Low | 1(ref) |  | 1 (ref) |  | 1 (ref) |  | 1 (ref) |  |
| Intermediate | 2.40 (2.12-2.71) | <0.001 | 1.87 (1.60-2.17) | <0.001 | 1.83 (1.58-2.13) | <0.001 | 1.35 (1.16-1.58) | <0.001 |
| High | 3.34 (2.86-3.88) | <0.001 | 2.28 (1.89-2.73) | <0.001 | 2.17 (1.81-2.61) | <0.001 | 1.32 (1.08-1.61) | 0.006 |

Logistic regression analysis was performed.

Crude: Unadjust model. Model1: adjusted for sex and age. Model 2: adjusted for model1 + current smoking, high risk alcohol consumption, education, house income, aerobic exercise and resistance training. Model 3: adjusted for model2 + diabetes, HbA1c, HOMA-IR.

## Discussion

In this study, we found the close association between higher OSA risk and atheroscelrogenic risk in the Korean population from nationwide cross-sectional study based on KNHANES data. Intermediate or high OSA risk assessed by the STOP-BANG score was associated with a 1.3-fold increased risk of atherogenesis (AIP > 0.24), and these associations remained robust even after adjusting for multiple confounding variables, particularly among those with diabetes.

The OSA risk calculated using the STOP-BANG questionnaire is associated with diabetes and metabolic syndrome. Previously, two studies that used KNHANES 2019–2020 reported that patients with diabetes mellitus, hypertension, and obesity have a higher risk of OSA, which increased synergistically when they concurrently had two or more of these conditions [26] and that a high STOP-BANG score is related to a higher risk of insulin resistance and metabolic syndrome [27]. Despite this, OSA risk is often inadequately assessed in clinical practice. The STOP-BANG questionnaire is simple and

**Table 3. Multivariate logistic regression analysis of presence of diabetes for high AIP(>0.24).**

| | Crude | | Model 1 | | Model 2 | | Model 3 | |
|---|---|---|---|---|---|---|---|---|
| | OR(95% CI) | P-value | OR(95% CI) | P-value | OR(95% CI) | P-value | OR(95% CI) | P-value |
| normal | 1(ref) | | 1 (ref) | | 1 (ref) | | 1 (ref) | |
| Prediabetes | 2.19 (2.05-2.71) | <0.001 | 2.26 (1.97-2.59) | <0.001 | 2.28 (1.98-2.62) | <0.001 | 1.67 (1.44-1.93) | <0.001 |
| Diabetes | 3.64 (2.75-3.91) | <0.001 | 3.70 (3.10-4.43) | <0.001 | 3.57 (2.98-4.28) | <0.001 | 1.49 (1.14-1.94) | 0.003 |

Crude: Unadjust model. Model1: adjusted for sex and age. Model 2: adjusted for model1+current smoking, high risk alcohol consumption, education, house income, aerobic exercise and resistance training. Model 3: adjusted for model2+HbA1c, HOMA-IR.

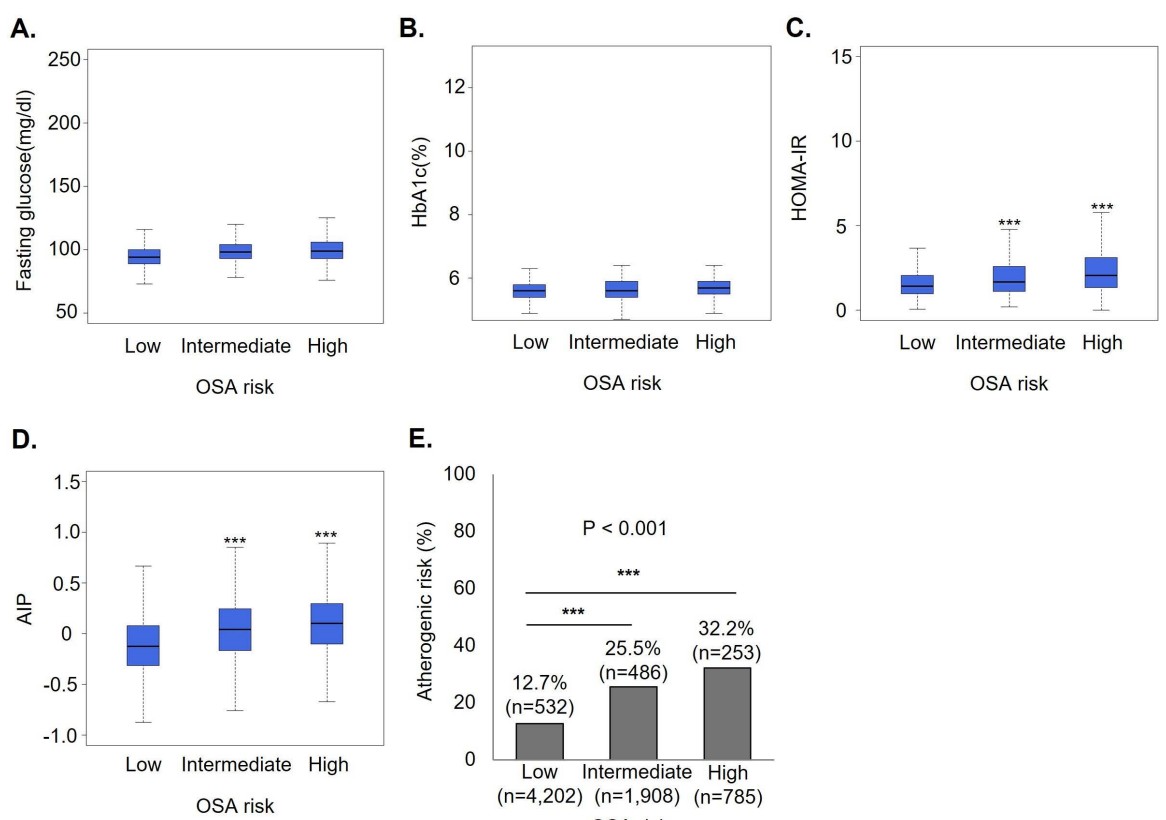

**Fig 3. Comparison of metabolic parameters and AIP based on OSA risk in populations without diabetes.**

can be quickly completed by patients without specialized knowledge. Its ease of use makes it a practical tool for routine screening, highlighting its clinical usefulness in improving early detection and management of OSA, thereby enhancing the management of patients' metabolic risks.

AIP is used to assess the risk of atherosclerosis and CVDs. AIP is easily obtainable and based on lipid profiles, including TG and HDL-C. Although the definite mechanism of the relationship between OSA and lipid profile has not been established, it is suggested that chronic intermittent hypoxia may elevate TG levels by enhancing enzymes and proteins involved in TG production in the liver [28]. Furthermore, chronic intermittent hypoxemia is regarded as the primary

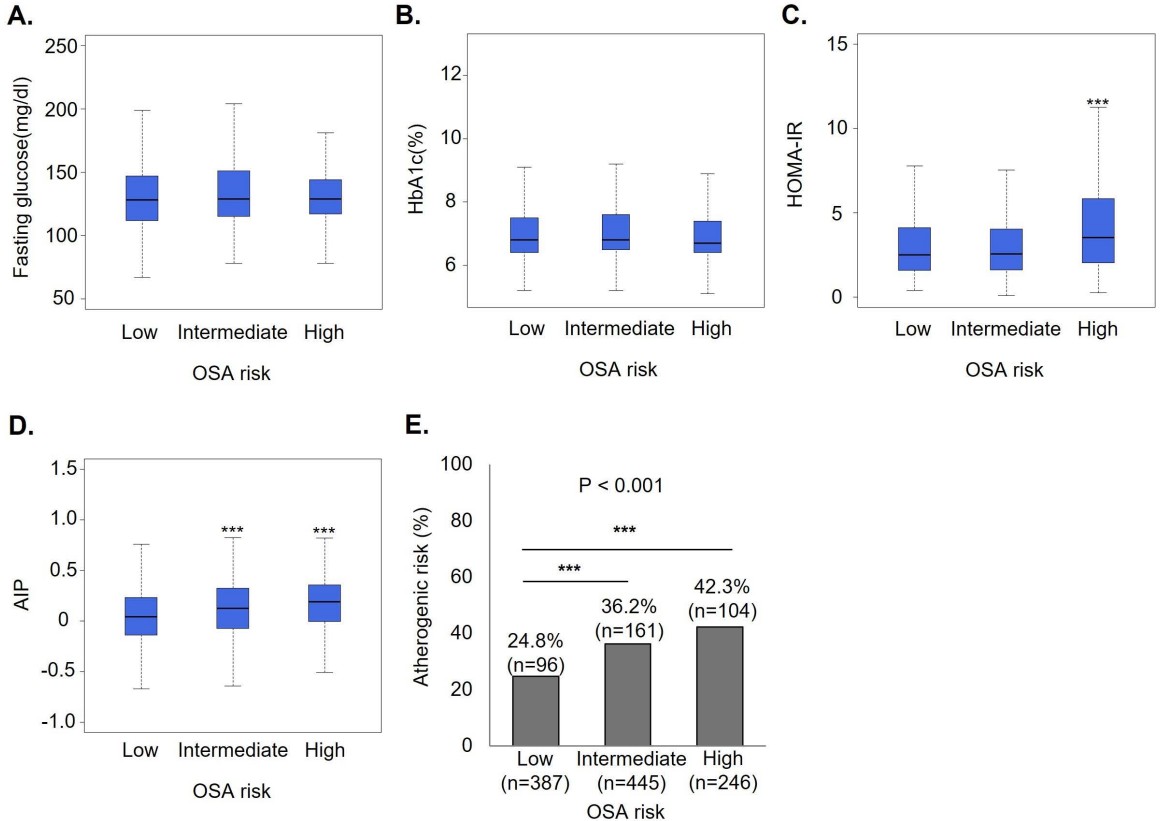

**Fig 4. Comparison of metabolic parameters and AIP based on OSA risk in populations with diabetes.** Comparison of intermediate or high OSA risk to low OSA risk: *p<0.05, **p<0.01, ***p<0.001. AIP, atherogenic index of plasma; OSA, obstructive sleep apnea.

mechanism that contributes to IR in OSA [29]. Previous studies have reported that the AIP value increases with increasing cardiovascular risk and is a superior predictor of CVD than individual lipid risk factors [10,24,30]. Another cross-sectional study involving 560 patients reported that OSA severity was independently associated with AIP, regardless of obesity, and that AIP demonstrated superior predictive value for nocturnal hypoxemia than individual lipid risk factors [31].

We investigated the relationship between the risk of OSA and AIP with respect to the presence of diabetes, as diabetes is a significant variable that increases the risk of atherosclerosis. Furthermore, a short sleep duration increases the risk of developing type 2 diabetes [32]. OSA contributes to diabetes through intermittent hypoxia, increased sympathetic activity, and elevated inflammation, which impair glucose homeostasis and insulin sensitivity [33]. In a previous study that included patients with type 2 diabetes, a STOP-BANG score ≥5 was associated with increased insulin resistance and was a predictor of 10-year fatal and non-fatal coronary heart disease risk [34]. In another cross-sectional study, a high risk of OSA (STOP-BANG score ≥5) showed a positive correlation with mean HbA1c and fasting glucose levels in patients with diabetes [35]. In our study, there was no significant difference in HbA1c and fasting glucose according to the STOP-BANG score among patients with diabetes. However, we observed a significant increase in HOMA-IR and AIP as the risk of OSA increased. These findings suggest that a high risk of OSA does lead to a sleep disorder, a complex condition with profound implications for cardiovascular health. Even among patients with diabetes with similar blood glucose levels, an increased OSA risk is associated with higher cardiometabolic dysfunction. Therefore, the STOP-BANG may be used as a simple predictor of atherogenic risk in diabetes care settings.

**Table 4. Effect of OSA on atherogenic risk (AIP > 0.24) according to the presence of diabetes.**

| (A) Normal + Prediabetes | Crude | | Model 1 | | Model 2 | | Model 3 | |
|---|---|---|---|---|---|---|---|---|
| | OR(95% CI) | P | OR(95% CI) | P | OR(95% CI) | P | OR(95% CI) | P |
| OSA risk | | | | | | | | |
| Low | 1(ref) | | 1 (ref) | | 1 (ref) | | 1 (ref) | |
| Intermediate | 2.36 (2.06-2.71) | <0.001 | 1.76 (1.48-2.08) | <0.001 | 1.73 (1.46-2.06) | <0.001 | 1.21 (1.02-1.45) | 0.03 |
| High | 3.28 (2.75-3.91) | <0.001 | 2.10 (1.71-2.59) | <0.001 | 2.04 (1.65-2.52) | <0.001 | 1.17 (0.93-1.47) | 0.17 |
| (B) Diabetes | Crude | | Model 1 | | Model 2 | | Model 3 | |
| | OR(95% CI) | P | OR(95% CI) | P | OR(95% CI) | P | OR(95% CI) | P |
| OSA risk | | | | | | | | |
| Low | 1(ref) | | 1 (ref) | | 1 (ref) | | 1 (ref) | |
| Intermediate | 1.72 (1.27-2.32) | <0.001 | 1.65 (1.15-2.36) | <0.001 | 1.62 (1.13-2.32) | 0.009 | 1.51 (1.05-2.19) | 0.03 |
| High | 2.22 (1.58-3.13) | <0.001 | 1.82 (1.20-2.76) | <0.001 | 1.74 (1.14-2.66) | 0.01 | 1.58 (1.02-2.46) | 0.04 |

Crude: Unadjust model. Model1: adjusted for sex and age. Model 2: adjusted for model1 + current smoking, high risk alcohol consumption, education, house income, aerobic exercise and resistance training. Model 3: adjusted for model2 + HbA1c, HOMA-IR.

This study has several limitations. First, the absence of polysomnography results, which are the gold standard for diagnosing OSA, limits the precision of our findings. The STOP-BANG score is known for its low specificity despite having a high sensitivity of up to 92% to 100% for severe OSA [8]. This means that the STOP-BANG score potentially inflates the associations observed between OSA and metabolic disorders. However, polysomnography is not feasible for large-scale studies of the general population owing to its high cost and limited accessibility [7,8]. In previous studies based on polysomnography, the sample sizes were relatively small, and participants were likely individuals already aware of their symptoms [25,31,36]. This suggests the possibility of selection bias, with more severe cases of OSA being overrepresented. Second, because this is a cross-sectional study, our research has limited ability to establish causality. Longitudinal studies with follow-up data are needed to better understand the directionality and causative pathways underlying these associations.

Despite these limitations, the strength of the current study is that it uses the STOP-BANG score to assess a large sample of the Korean general population. Our study highlights important associations between high STOP-BANG scores and atherogenic risk, underscoring the need for further research and integrated clinical approaches to address both sleep health and metabolic conditions.

The implications of these findings are significant. Our research highlights the importance of considering OSA, in the context of atherogenic risk. Clinicians should be aware that patients with high STOP-BANG scores may be at increased risk for atherogenic disturbances and should consider early interventions to manage these risks, potentially reducing the risk of CVD and other related complications.

In conclusion, OSA is linked to an increased atherogenic risk, particularly in patients with diabetes. Screening for high-risk OSA using the STOP-BANG might be beneficial in the diabetes care setting to prevent cardiometabolic complications. Further prospective studies that could provide deeper insights into the underlying mechanisms and potential benefits of OSA treatment in reducing atherogenic risk are warranted.

## Author contributions

**Conceptualization:** Il Rae Park, Seung Min Chung.

**Data curation:** Il Rae Park.

**Investigation:** Il Rae Park, Yong Geun Chung.

**Methodology:** Il Rae Park, Yong Geun Chung.

**Supervision:** Seung Min Chung.

**Visualization:** Il Rae Park.

**Writing – original draft:** Il Rae Park, Seung Min Chung.

**Writing – review & editing:** Il Rae Park, Yong Geun Chung, Seung Min Chung, Jun Sung Moon, Ji Sung Yoon, Kyu Chang Won.

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
