## [Decision Letter · Decision Letter 0]

9 Dec 2024

PONE-D-24-46140Association between Obstructive Sleep Apnea Risk and Atherosclerosis: A Nationwide Cross-sectional Study in the Korean PopulationPLOS ONE

Dear Dr. Chung,

Thank you for submitting your manuscript to PLOS ONE. After careful consideration, we feel that it has merit but does not fully meet PLOS ONE’s publication criteria as it currently stands. Therefore, we invite you to submit a revised version of the manuscript that addresses the points raised during the review process.

We look forward to receiving your revised manuscript.

Kind regards,

Qian Wu

Academic Editor

PLOS ONE

Journal Requirements:

**Additional Editor Comments:**

Please make peer-to-peer modifications to the reviewer's comments.

Reviewers' comments:

Reviewer's Responses to Questions

**Comments to the Author**

1. Is the manuscript technically sound, and do the data support the conclusions?

Reviewer #1: Yes

Reviewer #2: Yes

2. Has the statistical analysis been performed appropriately and rigorously? 

Reviewer #1: Yes

Reviewer #2: Yes

3. Have the authors made all data underlying the findings in their manuscript fully available?

Reviewer #1: Yes

Reviewer #2: Yes

4. Is the manuscript presented in an intelligible fashion and written in standard English?

Reviewer #1: Yes

Reviewer #2: Yes

5. Review Comments to the Author

Reviewer #1: The article is sounds intersting and up to the mark and statistical data is also ok but one revision i want to suggest you which is you used too old references like one is 1981 and some reference are 2001 or before 2015 try to use the references like from 2010 to 2024.. I am recoomending this with minor revision.

Reviewer #2: Few modifications have been suggested.

1. Page no. 7, line no. 138-to mention accreditation status of laboratories where tests were conducted

2. Page no. 7, line no. 145-reference is needed

3. line no. 152-is it operational definition?

4. Page no.10, line no. 208-to mention rationality of conducting non-parametric tests

6. PLOS authors have the option to publish the peer review history of their article (what does this mean? ). If published, this will include your full peer review and any attached files.

**Do you want your identity to be public for this peer review?** For information about this choice, including consent withdrawal, please see our Privacy Policy .

Reviewer #1: No

Reviewer #2: **Yes: ** Dr. Satabdi Mitra

---

## [Author Response · Author response to Decision Letter 1]

13 Jan 2025

Reviewer #1: The article is sounds intersting and up to the mark and statistical data is also ok but one revision i want to suggest you which is you used too old references like one is 1981 and some reference are 2001 or before 2015 try to use the references like from 2010 to 2024.. I am recoomending this with minor revision.

Thank you for your comments.

As you suggested, we have removed references published before 2010 and replaced them with more recent studies, ensuring the references from 2010 to 2024. The updated references are recorded in the revised manuscript for your review.

However, we have retained the following pre-2010 references due to their foundational significance in the field: the 2002 study by Kapur, V. et al., which provides a comprehensive analysis of the underestimation of sleep apnea prevalence; the 2008 study by Chung, F. et al., introducing the concept of the STOP-BANG score; and the 2006 study by Dobiásová, M. et al., which presents the concept of AIP.

We appreciate your understanding and thank you for your valuable suggestions.

Reviewer #2: Few modifications have been suggested.

1. Page no. 7, line no. 138-to mention accreditation status of laboratories where tests were conducted

2. Page no. 7, line no. 145-reference is needed

3. line no. 152-is it operational definition?

4. Page no.10, line no. 208-to mention rationality of conducting non-parametric tests

Thank you for your comments.

1. KNHANES provides various standards and regulations to ensure quality control of laboratory tests. This information has been included in the revised manuscript.

2. The definitions of the variables analyzed in this study are referenced from KNHANES. Therefore, KNHANES has been cited as a reference.

3. The definition of aerobic exercise is also referenced from KNHANES. Additionally, this definition aligns with the National Physical Activity Guidelines published by the U.S. Department of Health and Human Services.

4. The rationale for conducting non-parametric tests has been mentioned in the revised manuscript.

We appreciate your valuable suggestions.

---

## [Decision Letter · Decision Letter 1]

30 Mar 2025

Association between Obstructive Sleep Apnea Risk and Atherosclerosis: A Nationwide Cross-sectional Study in the Korean Population

PONE-D-24-46140R1

Dear Dr. Chung,

We’re pleased to inform you that your manuscript has been judged scientifically suitable for publication and will be formally accepted for publication once it meets all outstanding technical requirements.

Kind regards,

Qian Wu

Academic Editor

PLOS ONE

Additional Editor Comments (optional):

Reviewers' comments:

Reviewer's Responses to Questions

**Comments to the Author**

1. If the authors have adequately addressed your comments raised in a previous round of review and you feel that this manuscript is now acceptable for publication, you may indicate that here to bypass the “Comments to the Author” section, enter your conflict of interest statement in the “Confidential to Editor” section, and submit your "Accept" recommendation.

Reviewer #1: All comments have been addressed

Reviewer #2: All comments have been addressed

2. Is the manuscript technically sound, and do the data support the conclusions?

Reviewer #1: Yes

Reviewer #2: Yes

3. Has the statistical analysis been performed appropriately and rigorously? 

Reviewer #1: Yes

Reviewer #2: Yes

4. Have the authors made all data underlying the findings in their manuscript fully available?

Reviewer #1: Yes

Reviewer #2: Yes

5. Is the manuscript presented in an intelligible fashion and written in standard English?

Reviewer #1: Yes

Reviewer #2: Yes

6. Review Comments to the Author

Reviewer #1: Respected Author, All the comments which i asked you addressed them in intelligeable way while on the hand i can understand there is still one reference 2002 but now its okay sometime I understand the relevant data is not avaialable. I accept this manuscript.

Reviewer #2: Thank you authors for addressing such an important topic and answering all the review questions. The write up is really nice and all the statistical analysis have been well presented. The queries as raised in reviews have also been taken care of, answered and the manuscript has been modified accordingly and as needed. In the area context as well as in current global context it is an important health issue which is very often not adequately addressed. It needs more exploration in other areas in future researches. Authors may think to conduct the study on this topic over various other nations to get an overall picture which may help policy makers to chalk out health programmes on this issue with applicability in various countries.

7. PLOS authors have the option to publish the peer review history of their article (what does this mean? ). If published, this will include your full peer review and any attached files.

**Do you want your identity to be public for this peer review?** For information about this choice, including consent withdrawal, please see our Privacy Policy .

Reviewer #1: **Yes: ** Sammra Maqsood

Reviewer #2: **Yes: ** Dr Satabdi Mitra

---

## [Editor Report · Acceptance letter]

PONE-D-24-46140R1

PLOS ONE

Dear Dr. Chung,

I'm pleased to inform you that your manuscript has been deemed suitable for publication in PLOS ONE. Congratulations! Your manuscript is now being handed over to our production team.

Kind regards,

on behalf of

Dr. Qian Wu

Academic Editor

PLOS ONE